# Quality of maternal healthcare and travel time influence birthing service utilisation in Ghanaian health facilities: a geographical analysis of routine health data

Winfred Dotse-Gborgbortsi  ,[1] Andrew J Tatem,[1] Zoe Matthews,[2] Victor A Alegana,[3] Anthony Ofosu,[4] Jim A Wright[1]

[1]School of Geography and Environmental Science, University of Southampton, Southampton, UK
[2]Department of Social Statistics and Demography, University of Southampton, Southampton, UK
[3]Population Health Unit-Wellcome Trust Research Programme, Kenya Medical Research Institute, Nairobi, Kenya
[4]Headquarters, Ghana Health Service, Accra, Greater Accra, Ghana

**Correspondence to**
Winfred Dotse-Gborgbortsi;
w.w.dotse-gborgbortsi@soton.
ac.uk

## ABSTRACT

**Objectives** To investigate how the quality of maternal health services and travel times to health facilities affect birthing service utilisation in Eastern Region, Ghana.
**Design** The study is a cross-sectional spatial interaction analysis of birth service utilisation patterns. Routine birth data were spatially linked to quality care, service demand and travel time data.
**Setting** 131 Health facilities (public, private and faith-based) in 33 districts in Eastern Region, Ghana.
**Participants** Women who gave birth in health facilities in the Eastern Region, Ghana in 2017.
**Outcome measures** The count of women giving birth, the quality of birthing care services and the geographic coverage of birthing care services.
**Results** As travel time from women's place of residence to the health facility increased up to two2 hours, the utilisation rate markedly decreased. Higher quality of maternal health services has a larger, positive effect on utilisation rates than service proximity. The quality of maternal health services was higher in hospitals than in primary care facilities. Most women (88.6%) travelling via mechanised transport were within two2 hours of any birthing service. The majority (56.2%) of women were beyond the two2 -hour threshold of critical comprehensive emergency obstetric and newborn care (CEmONC) services. Few CEmONC services were in urban centres, disadvantaging rural populations.
**Conclusions** To increase birthing service utilisation in Ghana, higher quality health facilities should be located closer to women, particularly in rural areas. Beyond Ghana, routinely collected birth records could be used to understand the interaction of service proximity and quality.

## INTRODUCTION

Substandard maternal health quality in some health facilities in low/middle-income countries (LMICs) has led to excess obstetric complications and maternal deaths.[1] Though quality maternal healthcare during labour is vital in reducing maternal mortality,[2] and despite the steady rise in skilled birth

## STRENGTHS AND LIMITATIONS OF THIS STUDY

⇒ We integrated disparate data sources to estimate access, quality and use of birthing services.
⇒ We extracted patients' place of residence and destination facilities from routine health records, unlike most studies which use residential locations only from one-off household surveys.
⇒ Analysis excluded women whose place of residence could not be located, which could lead to selection bias.
⇒ Due to data limitations, we could not account for individual sociodemographic characteristics that affect birthing service utilisation.

coverage, maternity services remain below acceptable levels in most of sub-Saharan Africa.[3] The rising rate of health facility births is not associated with a decline in maternal deaths due to substandard health facilities incapable of averting maternal deaths.[4] Indeed, an estimated 92.5% of sub-Saharan Africa's population is within 2 hours' journey time of a hospital,[5] but maternal mortality is indicative of poor care.

There is ample evidence on the determinants of health facility utilisation for birth and the causes of maternal mortality.[6 7] Increasing distance to the closest health facility decreases the odds of health facility births[8] and increases maternal mortality rates.[9] For referred cases between health facilities, the maternal health risk is higher for longer travel times.[10] Geographical accessibility hinders decisions to attend a health facility during pregnancy or seek care in a health facility for complications during home birth.[11] Furthermore, after deciding to use a health facility, proximity to care influences the choice of facility, journey time and transportation mode.[12] Therefore, to prevent maternal deaths, it

is recommended that quality health facilities be within 2 hours' journey time of residential locations.[13]

There is a paucity of studies using routine health data to evaluate the relationship between quality maternal healthcare and birthing service utilisation.[14] This is because the data can be incomplete, difficult to access, and expensive to extract and process.[15 16] Most studies conduct one-off subnational surveys to collect data[17 18] or rely on national household surveys such as the Demographic and Health Survey.[19 20] However, surveys can be expensive and infrequent. Despite the challenges with routine health data, it has become increasingly important for malaria and maternal health research in sub-Saharan Africa in recent years.[21]

When women overcome geographical barriers to use birthing services, the quality of maternal healthcare on offer could affect the timeliness and adequacy of care. A review of supply-side barriers limiting access to quality maternal healthcare found medicines and equipment, staff capacity and morale, infrastructure and referral systems as some of the challenges.[22] In addition, privacy during consultation and labour, cleanliness and well-kept physical surroundings were other factors that determined service satisfaction among women seeking birthing services.[23] Although the Community Health Planning and Services (CHPS) initiative aims to achieve universal health coverage in Ghana, there is substandard maternal care in some CHPS facilities.[24 25] Thus, women are likely to travel longer to receive care in a health facility perceived as better than the nearest one.[26]

There is inadequate evidence on healthcare utilisation and proximity to quality maternal care to support Ghana's maternal health interventions and programming. The latest analysis calculating journey times to different qualities of maternal health services is over a decade old.[27] Furthermore, the study measures potential access and did not estimate the effect of quality on utilisation rates. Two recent studies integrated quality care indicators estimated via service provision assessment (SPA) with secondary data analysis to predict the probability of skilled attendance at birth[28] and health facility births.[29] However, both studies estimated journeys to the nearest health facility, assuming women do not bypass health facilities to seek care elsewhere.[30] In the absence of detailed spatial information on where women reside and the actual health facilities they use, realised access modelling is challenging.

Building on previous work,[30] this study aims to determine the effect of quality maternal health services and travel time on birthing service utilisation via geographical accessibility analysis by integrating routine birth data, SPA and ancillary spatial data. Utilisation is modelled as a product of travel time from place of residence to health facilities, population demand in place of residence and maternal service quality in health facilities. Furthermore, the study aims to assess how different facility characteristics influence utilisation. Finally, it aims to estimate the number of women within 2 hours' walk or mechanised journey from any birthing service and from

comprehensive emergency obstetric and newborn care services (CEmONC).

## METHODS
### Research design and setting
The study is a cross-sectional spatial interaction analysis of birth service utilisation. Spatial interaction models predict the movement of people, goods or services from one point to another.[31] The gravity models in this study predict the number of women making trips from a place of residence to a health facility to give birth as a function of travel times, the population of women and the quality of maternal health services.

The study was conducted in Eastern Region, Ghana. The Eastern Region has 33 administrative districts divided into 225 subdistricts. The 2021 population census estimates 2.9 million persons in the region of which approximately 50.8% are females and 51.5% live in rural areas. There are 28 hospitals providing secondary care and 1136 primary health facilities (140 health centres, 78 clinics, 884 CHPS, 31 maternity homes, 3 polyclinics). The majority of the health facilities are public (89.4%) with some (8.8%) private. Since 2008, Ghana introduced a free maternal health policy that covers pregnant women giving birth in all public and some private health facilities.[32]

### Routine birth data
Eligible subjects were women with a facility-based birth record between 1 January and 31 December 2017. Women whose place of residence was missing or non-mappable were excluded from analysis. All public health facilities report aggregate counts of women using birthing services in the DHIMS (District Health Information Management Systems) but only secondary facilities capture individual women's birth records electronically in the Ghana Health Service's (GHS) managed DHIMS. For primary health facilities, birth data were extracted from manually written book registers using a data extraction form. In secondary facilities, birth data were downloaded from DHIMS. Patient flows were calculated between origins (women's place of residence) and destinations (health facility locations) in the routine birth data. Details of the health facility used and the place of residence obtained from the routine birth data were used to spatially link the demand population, travel times and quality of care metrics. The women's characteristics collected but not analysed are age, parity, level of education and occupation.

### Demand population
The number of women aged 15–45 years was estimated from WorldPop's 100 m resolution gridded age and sex-disaggregated population projections for 2020.[33] WorldPop develops the population estimates by disaggregating administrative unit-linked census data into building footprints using machine learning methods and a library of geospatial covariates. Satellite-derived building footprints ensure populations are assigned to

grid cells where people are known to live. We identified the population of women for each residential place by least cost travel time, then summed population counts for each place of residence to estimate demand within residential place catchments. GHS provided the list of place names with geographical coordinates.

Preliminary model fitting shows population demand disaggregated by place of residence better explained utilisation patterns than population demand at subdistrict level (online supplemental appendix 1). Demand was therefore modelled by place of residence.

## Maternal healthcare quality metrics

An SPA was conducted in August and September 2021 to collect data on health facility attributes. Health facilities averaging five births per month in 2017 were surveyed and their geographical coordinates collected—for spatial linkage—via mobile data collection software.[34] Out of the 1136 health facilities, 150 were eligible. However, 19 of the 150 eligible facilities were excluded because they could not provide individual-level routine birth register records for analysis. Thus, 131 health facilities were analysed.

Ten care quality domains were created from SPA data and combined to construct a quality of maternal healthcare composite index. The domains are human resource capacity, EmONC signal functions, medicines, nonmedical supplies, amenities and infrastructure, referral systems, staff morale, privacy, training, and Water, Sanitation and Hygiene (WASH).

Domains were normalised (0–1) using range standardisation to ensure none of the domains unduly influenced the summary composite index. Domain scores were unweighted as there is no published evidence of their relative effects on utilisation.[35] The 10-domain scores were averaged to derive the final composite score.[36] The internal consistency of the domains was verified with Cronbach's alpha. The 10 domains correlated well for analysis (Cronbach coefficient=0.74 (95% CI 0.65 to 0.80)). The composite index was grouped into quintiles.[37]

Additional indicators of maternal health quality were estimated and evaluated against the composite 10-dimension index. Two additional composite indices were derived from a principal component analysis. The first included human resource capacity, number of EmONC signal functions, medicines, non-medical supplies, amenities and infrastructure, and referral systems. The second component was based on staff morale, privacy, training and WASH.

## Travel time model

Topographical data were used to model journey times from place of residence to health facilities. Travel times were modelled as the least cost path over an impedance surface. An impedance surface is a gridded map layer depicting travel speed. Road networks and water bodies from OpenStreetMap (OSM), a global mapping platform,[38] were incorporated into this layer. We estimated travel by walking and multimodal (walking and motorised

journeys). The multimodal model combined walking on traversable land cover classes (mapped by the European Space Agency at 10×10 m, year 2020)[39] and motorised travel on roads. Walking speeds (online supplemental appendix 2) were adapted from previous studies using similar models.[40 41]

Motorised speeds on roads were inferred from Global Positioning System tracks collected via mobile devices. During the SPA data collection, we tracked road networks (767 km), and recorded their condition (good, bad, very bad) at the time of data collection. GHS drivers who frequently travel within the study area added similar tags (good, bad, very bad) to printed OSM maps. Subsequently, OSM volunteers digitised the paper maps, uploading these to the OSM platform. The average travel speeds were proportionally weighted by road class and condition (online supplemental appendix 3).

The land cover, roads and water bodies were combined into a gridded friction dataset value representing the feasibility of traversing the landscape. The Tobler function[42] and elevation data (Shuttle Radar Topographic Mission 30 m)[43 44] were used to model terrain effects on walking speed.

Travel times were estimated to health facilities providing any birthing service and secondary CEmONC facilities. Travel times were categorised to show critical thresholds for obstetric emergencies,[13 45] counting women living in each travel time zone with the census-derived WorldPop gridded population of women 15–45 years.

## Modelling birthing service utilisation

The gravity-type spatial interaction model (SIM) framework applied in this study follows the Wilson group of SIMs.[46] Spatial interaction models predict the movement of services, goods or persons between two locations.[31] Gravity models, based on Newton's gravitational law, use the mass of two objects and the distance between them to predict their spatial interaction flows. Unconstrained, origin-constrained and destination-constrained models were implemented in this study. The unconstrained model fits coefficient for facility quality, proximity and population demand. In contrast, the origin-constrained model replaces the population demand term with separate coefficients for each origin, while the destination-constrained model replaces facility attractiveness with separate coefficients for each destination.

Travel time, population demand and care quality predictors were included in the models due to their influence on birthing service utilisation. In LMICs, systematic review evidence shows that proximity to a health facility significantly increased facility-based births.[47] The population of women was included because more women of childbearing age living near a health facility should increase births. Lastly, health service quality could affect birthing service utilisation at nearby health facilities.[29] We hypothesise that staff morale, privacy, training and WASH quality dimensions would have a greater effect on birthing service utilisation than human resource capacity,

number of EmONC signal functions provided, availability of medicines, non-medical supplies, amenities and infrastructure, and referral systems.

The predicted outcome is the number of women making trips from a residential community to a health facility. The data are a patient flow matrix between all residential places and health facilities. Therefore, there were zeros where no interactions occurred between a residential population and a health facility. Consequently, the Poisson model was overdispersed (8.2, z=6.35, p<0.001) and zero inflated. Hence, a zero-inflated, negative binomial model was implemented to address overdispersion and excess zeros.[48 49] Travel time between place of origin and destination facility was used to predict excess zeros because women would usually not travel unreasonably long distances to give birth in a health facility. The independent variables were log transformed for linearity to facilitate SIM calibration in a regression framework.[50] To enhance interpretation, we relied on marginal predicted counts to estimate utilisation and the interaction between quality and travel time from the unconstrained model. Likewise, we standardised the model estimates for easier interpretation. The incidence rate ratios (IRRs) of the constrained models were mapped. The Akaike Information Criterion (AIC) was used to evaluate the relative predictive importance of the quality care metrics and population demand estimates. The AIC was used to evaluate models.[51] The coefficient of determination and root mean square error were used to evaluate the final gravity models.

### Patient and public involvement

This research was done without patient involvement. Patients were not invited to comment on the study design and were not consulted to develop patient-relevant outcomes or interpret the results. Patients were not invited to contribute to the writing or editing of this document for readability or accuracy.

## RESULTS

There were 40 911 women from 964 places of residence who gave birth in 131 health facilities included in the analysis. The majority (75.6%) gave birth in secondary facilities. Aggregate reports in DHIMS recorded births by 57 018 women, of whom 47 900 had corresponding individual records (DHIMS: 70%, paper register: 30%). Of these individual records, 42 205 were geocoded (DHIMS: 73%, paper register: 27%).

### Quality of maternal healthcare and demand for birthing services

Figure 1A shows the geographical distribution of women 15–45 years in the study area, representing demand for birthing services. There were 2000 or fewer women at most origins (84.1% of 964). Most districts had at least one highly populated place of residence, with an above average number of female residents. There were few (17

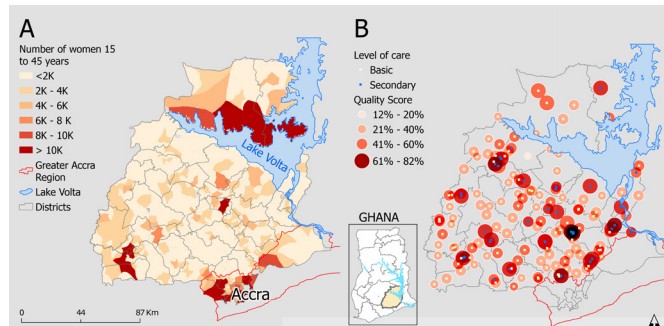

**Figure 1** (A) Geographical distribution of women 15–45 years by place of residence (origins). (B) Spatial distribution of health facilities by quality and level of care.

(1.82%)) highly populated origins, mostly in Accra, with at least 10 000 women.

The average quality score for health facilities was 48%. The quality scores ranged from 12% to 82%. There was a 20% quality difference between the mean scores at primary (43.4%) and secondary health facilities (63.6%). Among primary health facilities, CHPS (39.6%) and maternity homes (37.8%) had the lowest average quality compared with health centres (44.7%) and polyclinics (45%). Approximately five health facilities per district provided birthing services, and the majority were health centres. Figure 1B shows lower quality scores in primary health facilities and clustering of secondary health facilities in some urban areas.

### Modelled travel time and population access to care

Figure 2 shows the spatial inequalities in geographical access to birthing services. Most settlements could reach any form of birthing service within 2 hours (figure 2A). However, only a few settlements could reach a higher quality CEmONC health facility capable of handling complications within that same time (figure 2B). There is thus inadequate provision of CEmONC services, with most areas beyond 2 hours' travel. Travel times were estimated for all 853 085 female residents aged 15–45 years within Eastern Region. Over 50% of these women were within an hour's travel to a health facility offering any birthing service (figure 2C). In contrast, most women had more than 2 hours' travel from CEmONC facilities providing lifesaving services such as blood transfusion and caesarean sections (figure 2D). Less than 35% of women could walk or travel by mechanised transport to the nearest secondary care health facility with CEmONC services within the recommended 2-hour threshold.

### Evaluating the influences on birthing service utilisation

The summary quality index calculated from the 10 domains explained birthing service utilisation better than the other quality care indicators. Routine services comprising staff motivation, privacy, training and WASH had the least effect on birthing service use (online supplemental appendix 4).

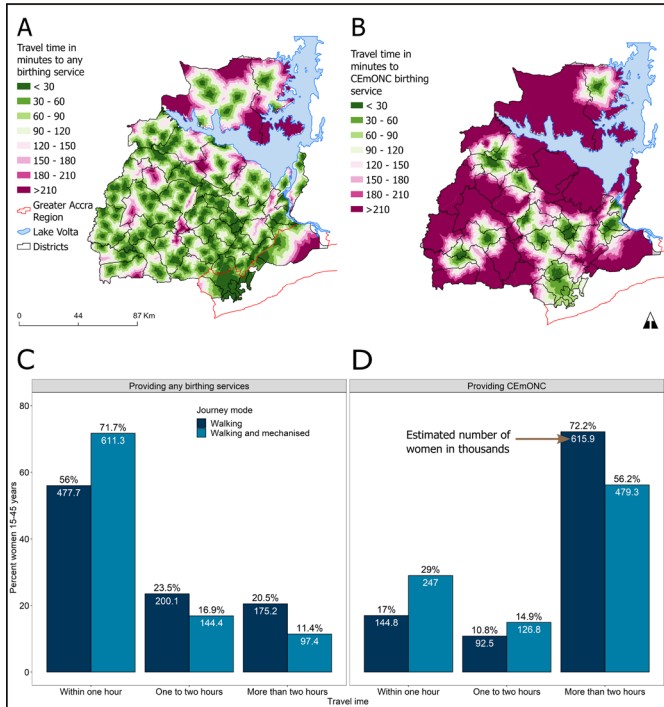

**Figure 2** Geographical distribution of multimodal (walking plus mechanised) travel time to (A) any birthing service and (B) health facilities offering CEmONC; per cent and number of women 15–45 years living within walking and mechanised travel time thresholds to (C) any birthing service and (D) CEmONC health facilities (n=853 085). CEmONC, comprehensive emergency obstetric and newborn care.

The coefficients for the unconstrained model in table 1 were transformed into IRR and standardised for easier interpretability. Higher travel time to birthing services decreased the count of women attending to give birth. Model results in table 1 show that the count of women using birthing services decreased by 57.6% (IRR 0.3, 95% CI: 0.28 to 0.32) per SD increase in travel time. In contrast, population demand for services and quality of care increased health service utilisation. Number of women attending for childbirth increased by 36.5% (IRR 1.3, 95% CI: 1.23 to 1.36) when the SD of population demand increased by a unit. The effect of quality care on utilisation was higher than travel time and population. There was a 208.3% (IRR 140.19, 95% CI: 109.39 to 179.68) increase in the count of women giving birth per unit increase in the SD of the care quality index.

The destination-constrained model had the highest correlation between predicted and actual counts of women ($R^2$: 10.6%). Although the unconstrained model reported the lowest correlation (5.5%), it had the lowest root mean square error (23.3). The origin-constrained model had the lowest AIC (table 1).

The average marginal effects, calculated from the unconstrained model, are presented in figure 3. The estimates show a profound decrease in service utilisation within the first 2 hours' travel to facility, where utilisation drops from 0.073 (95% CI: 0.067 to 0.079) at 1 hour to 0.030 (95% CI: 0.028 to 0.033) at 2 hours (figure 3A). Thereafter, the change in the predicted number of women was marginal from 3 hours and beyond. Service utilisation gradually increases with population demand for services and then plateaus (figure 3B). Greater quality of maternal healthcare increased utilisation exponentially (figure 3C). For a 10% increase in quality from 40% to 50%, the marginal effect doubles from 1.51 (95% CI: 1.19 to 1.83) to 2.47 (95% CI: 1.91 to 3.04) women, respectively. From 50% to 100% quality, the effect on utilisation quadruples to 10.91 (95% CI: 7.80 to 14.03). Given the same quality, the number of predicted women is significantly higher at lower travel times as shown in the interaction marginal effects plot (figure 3D). Furthermore, the interactions do not have a significant effect beyond the critical 2-hour threshold.

**Table 1** Incidence rate ratio (IRR) estimates from a zero-inflated negative binomial model predicting number of women using birthing services (n=40 911)

| Model | Unconstrained IRR (95% CI) | Origin constrained IRR (95% CI) | Destination constrained IRR (95% CI) |
|---|---|---|---|
| Intercept | 91.1 (75.74 to 109.57) | 9.98 (0.72 to 137.67) | 1.79 (0.93 to 3.44) |
| Log travel time in hours | 0.30 (0.28 to 0.32) | 0.16 (0.15 to 0.18) | 0.21 (0.2 to 0.23) |
| Log number of women 15–45 years | 1.30 (1.24 to 1.36) | – | 1.26 (1.2 to 1.32) |
| Log quality care | 140.20 (109.39 to 179.68) | 703.02 (541.06 to 913.46) | – |
| Intercept | 0.18 (0.14 to 0.23) | 0.14 (0.1 to 0.18) | 0.08 (0.06 to 0.1) |
| Log travel time | 6.88 (6.19 to 7.65) | 5.91 (5.24 to 6.68) | 7.82 (6.87 to 8.9) |
| Model evaluation | | | |
| AIC | 29 444 | 27 446 | 27 765 |
| RMSE* | 23.3 | 357 | 147 |
| $R^2$ (%)* | 5.5 | 8.1 | 10.6 |

*RMSE and $R^2$ were estimated with the observed versus fitted number of women giving birth.
AIC, Akaike Information Criterion; RMSE, root mean square error.

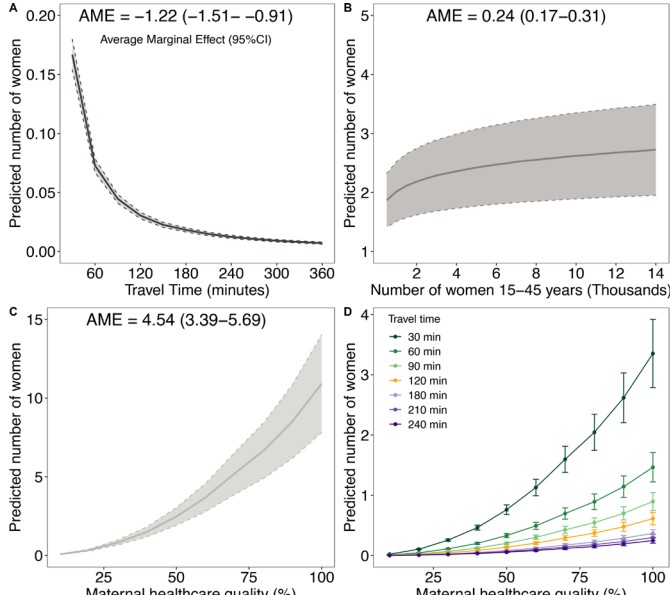

**Figure 3** Predicted marginal effects on birthing service utilisation for (A) travel time, (B) demand for services and (C) quality of maternal health services, (D) interacting quality and travel time effect based on the unconstrained model.

Higher residential population increased utilisation rate ratios. However, effects for most places of residence (534 (58.1%)) did not significantly influence birthing service utilisation. Residential locations with lower rate ratio estimates were mostly insignificant, in contrast to residential locations with higher rate ratios (figure 4A). Furthermore, SEs for estimates were higher at the residential locations with lower estimates.

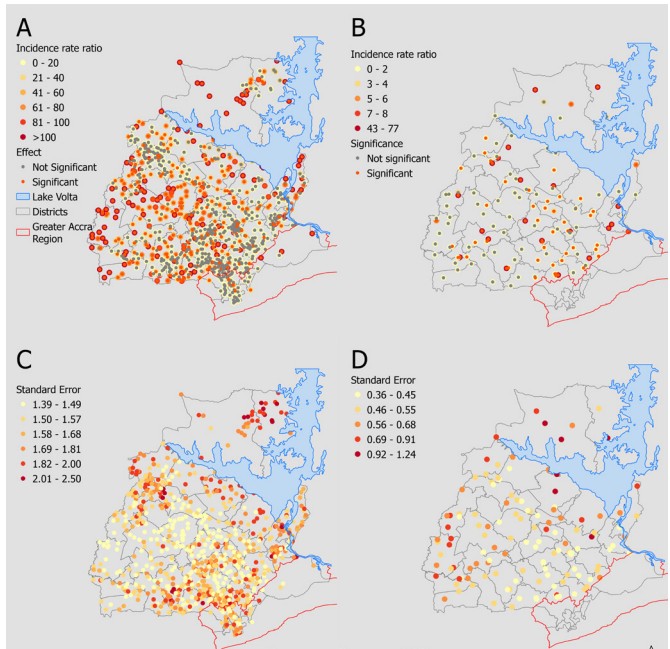

**Figure 4** Estimated incidence rate ratio effects at (A) residential town and (B) health facilities on the number of women using birthing services. SE of estimates at (C) residential town and (D) health facilities.

The destination-constrained model effects for health facilities are mapped in figure 4B. The reference health facility had the median quality score (47%) and is a health centre. Compared with the reference health centre, CHPS facilities with relatively lower quality scores reduced the rate ratio of utilisation by 0.57 (95% CI: 0.17 to 2.25), while a 1.35 (95% CI: 0.38 to 5.43) increase was observed at health centres. Hospitals increased the utilisation rate ratio by a factor of 16.15 (95% CI: 7.18 to 36.56). Just over half the health facilities (51.5%) had a statistically significant influence on utilisation. Of 30 hospitals, 25 (83.3%) significantly influenced birthing service utilisation compared with 42% of the 100 primary health facilities. There was relatively higher error at locations with lower incidence rates and vice versa (figure 4C,D).

## DISCUSSION

The quality of maternal health services in health facilities had a greater effect on the use of birthing services than geographical accessibility. Including additional quality care dimensions incrementally explained more of the variation in birthing service utilisation There was a significant inverse effect between travel time and birthing service utilisation, while increased population demand and quality of maternal health services promoted utilisation. Furthermore, many women lived beyond the WHO-recommended 2-hour travel time thresholds to CEmONC health facilities.[13 45]

While most studies analysing the use of birthing services in health facilities have relied on cross-sectional surveys,[29 52] only one recent study used routine Health Management Information Systems (HMIS) data collected on an ongoing basis.[30] Relative to this earlier study, we develop the use of routine HMIS data by analysing patient flows from primary and secondary care levels with updated SPA and improved geographical accessibility estimates. Also, the earlier study calibrated an unconstrained gravity model only, whereas the current study estimates effects for individual origins and destinations.

### Quality care effect and implications for maternal health

The findings in this study are consistent with the limited existing literature assessing healthcare quality and proximity's impacts on birthing service utilisation. In five African countries, maternal care quality was lower in primary health facilities than in secondary care facilities and higher quality was associated with higher utilisation rates.[53] In these five countries, standard linear regression modelling was used to predict quality care from birth volume, country, skilled staff per bed capacity and health facility ownership (public/private).

There is a lack of gravity-type SIMs in low/middle-income settings to facilitate methodological and result comparisons. Two studies in Ghana did not find a significant association between maternal care quality and health service utilisation.[29 30] Nesbitt *et al* modelled utilisation to the nearest health facility. However, there is a

high bypassing rate of health facilities in Ghana.[30] Nesbitt *et al*[29] argued that poor maternal health service quality metrics and homogeneity in service quality among health facilities could result in a weak relationship between health facility quality and utilisation. The other study did not observe significant variations in quality because they assessed only hospitals.

This study shows that health facilities with higher quality were associated with higher utilisation rates. A systematic review identifies characteristics of health facilities that satisfy women when they seek maternal healthcare in LMICs.[23] They found that women express higher satisfaction with maternal health services when there is sufficient infrastructure such as electricity, water and adequate bed capacity. Furthermore, the number of staff and their availability, particularly when women seek emergency obstetric care, determines service satisfaction.[54] Another critical factor for women is privacy during consultation, labour and after birth.[55]

Higher level of health facilities (primary/secondary) and health facility type (CHPS, health centre, hospital, etc) were associated with significantly greater utilisation, as were health facility capacity, routine care and EmONC signal functions. However, a comprehensive 10-domain quality index explained utilisation better than these simpler measures, suggesting improving staff morale, privacy, WASH and other health facility quality characteristics may also incrementally increase service utilisation. The SPA data collection tool was constructed with questions from different maternal health quality care tools to derive a broader composite index.[56] Therefore, our quality care index composition is essential because inclusion of additional quality components incrementally explained more variation in birthing service utilisation (see online supplemental appendix 4).

Higher quality care promotes maternal health service utilisation but does not always lead to the desired health outcomes.[57] For example, higher volumes of health facility births did not reduce maternal and perinatal deaths in Ghana, but stillbirths were lower at facilities with improved quality.[4] Hence, as efforts continue to increase skilled attendance at birth, health facilities should be prepared to manage complications to prevent maternal deaths.

### Proximity effect and implications for maternal health

Although our current study did not quantify bypassing of health facilities, the utilisation patterns were similar to our previous study,[30] suggesting a high level of bypassing. Poor women travelling from rural areas are most likely to bypass substandard health facilities for a hospital.[37] Furthermore, our map of health facilities shows the uneven distribution of secondary care facilities. While there are rural districts without hospitals, hospitals are clustered in urban areas, leading to unequal geographical access to CEmONC in the region.

The clustering of higher quality maternal care in urban areas implies that rural women in labour will travel further

and spend more to receive better care. Consequently, complications can result in death due to the longer travel times. Almost all the ambulances for transporting patients in emergencies are in these urban hospitals. Finally, the spatial distribution adds to the indirect costs of women travelling from rural to urban health facilities.[58]

Similar to our findings, several studies found geographical proximity to a facility increases birthing service utilisation.[59] Higher quality health facilities with CEmONC were more than 2 hours' travel away for most women. A study in 2012 shows approximately 63% women within 2 hours of CEmONC facilities in the study area, 19.1% higher than our estimate.[27] The change in CEmONC coverage could be due to expansion of geographical coverage of CEmONC services not matching population growth or a decline in the quality of secondary facilities.

The extent of the geographical coverage of any birthing service implies improved access for uncomplicated births, but there is a high risk for complicated ones. Hence, some women in obstetric emergencies living beyond the 2-hour critical threshold might die en route to a quality health facility with blood transfusion or surgical services.[13 60] The 2-hour recommend travel time[13] is relevant because bleeding is the leading direct cause of maternal mortality in sub-Saharan Africa.[61] Strategically upgrading some existing health facilities, particularly in rural areas, to provide CEmONC would reduce these inequalities in geographical access to quality healthcare. While there are calls for expanding access to essential obstetric care, the demand should be carefully considered to ensure these specialised services such as surgery and blood transfusion are not underused.[62]

The methods are transferable and scalable to settings with similar maternal health system structures, and data via health information systems. The findings are transferable to most regions in Ghana due to the similar health system structure. A key feature influencing healthcare-seeking travel by Ghanaian women is the unrestricted choice of health facilities,[30] variations in quality and the no cost policy.[32] In Ghana, the cost of giving birth in public facilities is free under a national health insurance scheme at a minimal registration cost.[32] Thus, women might choose the best health facility near them if they can afford travel and other costs. Countries with similar financial models are likely to observe comparable patterns and similar effects of proximity and the service quality on birthing service utilisation.

The analysis does not cover women giving birth in underused health facilities because birth data were extracted from health facilities averaging five births per month, consistent with previous EmONC SPAs.[63] Birth records were not available from 19 health facilities; the births in these facilities are 6.5% of total births, which poses some bias. Also, women whose places of residence could not be linked with a geographical coordinate were excluded. However, there was only a marginal difference in geocoding success rates by health facility ownership type, suggesting potential selection bias, as shown in

online supplemental appendix 5. Aggregated data versus individual and geocoded data were close to the line of agreement for most health facilities (online supplemental appendix 6), suggesting individual registers are complete. Furthermore, the individual birth register records analysed are substantially more complete than in the previous study and, unlike this earlier study, includes all health facility tiers.[30] Data quality can be improved by scaling up digitisation of individual birth data in DHIMS to all health facilities and improving the documentation of place names via reference datasets for geocoding to ensure uniform spatial scales. Since some women listed towns as their place of residence while others listed neighbourhoods within towns, the variable spatial precision of place names in urban areas might have affected model estimates. The study was also unable to account for individual characteristics that influence utilisation such as wealth and education.

The accuracy of the travel time estimates is contingent on the accuracy of input data such as road networks, travel speeds, land cover, elevation and water bodies. Besides, transient geographical barriers such as impassable roads due to broken bridges, flood inundation, road diversions and other challenges were not incorporated into the travel time estimates. In addition, transportation cost and availability can limit access to health facilities. Finally, modelled travel times often overestimate geographical access to care, relative to self-reported travel times from patient surveys.[64] We did not account for edge effects,[65] the tendency for women in Eastern Region to use birthing services in other regions or vice versa. Some women with longer travel times within Eastern Region could thus travel shorter distances to nearby regions.

Birthing service demand was estimated from gridded population datasets derived from the Ghana 2010 census and other datasets.[66] The population estimates are disaggregated into building extents where people live, but this can underestimate populations in urban areas.[67] The higher error associated with the utilisation rate ratios in urban areas could be due to the population estimates and the variable geocoding precision in these areas.

This study does not seek to infer causality from the cross-sectional design as the birth data, SPA and population estimates represent one period only.

## CONCLUSION AND FUTURE RESEARCH

Since health service quality is associated with greater birthing service utilisation, our study suggests that increasing service quality may drive up utilisation as well as improving some health outcomes. A 10-domain index incorporating service components such as staff morale and patient privacy better explained utilisation than facility capacity or range of clinical signal functions alone. Our finding implies these quality improvements could also lead to more modest utilisation increases. To increase the use of birthing services, higher quality health facilities should be located closer to women, particularly

in rural areas. Since this cross-sectional study relies on routine data which have improved in completeness and quality over time, in future, it should be possible to assess how changes in service quality or geographical coverage affect utilisation through spatiotemporal analysis of routine data. Having identified substantial bypassing in our analysis, in a future study, we plan to investigate facility bypassing as an outcome in relation to facility-level characteristics and the limited individual-level characteristics that are recorded on patient registers via a multilevel framework. Further analysis can include additional origin or destination characteristics such as health facility ownership or population ethnicity at origins.

**Acknowledgements** We are grateful to the Ghana Health Service for making their data available for analysis. Furthermore, we extend our gratitude to the midwives, health directors and health information officers who supported this study in different capacities. Solomon Boamah, Patient Dodge, Doris Mantey, Julius Gafli and James Otaniba Okoi assisted in the service provision assessment data collection. We express thanks to Ghana OpenStreetMap community, particularly Seth Enock Nyamador and Samuel Darkwah Manu, for updating the free spatial data used in modelling the travel times and the Ghana Health Service drivers who annotated road conditions on paper maps. We acknowledge the Economic and Social Research Council for funding this study through the South Coast Doctoral Training Partnership.

**Contributors** WD-G is the guarantor and accepts responsibility for the overall content. WD-G and JAW conceptualised and designed the study. WD-G analysed the data and wrote the original draft manuscript. JAW, AJT, ZM, VAA and AO supervised the analysis and reporting. All authors revised and edited the manuscript. All authors read and approved the final manuscript.

**Funding** The study was funded by the Economic and Social Research Council (ESRC), UK (grant number ES/P000673/1) through the South Coast Doctoral Training Partnership.

**Map disclaimer** The inclusion of any map (including the depiction of any boundaries therein), or of any geographic or locational reference, does not imply the expression of any opinion whatsoever on the part of BMJ concerning the legal status of any country, territory, jurisdiction or area or of its authorities. Any such expression remains solely that of the relevant source and is not endorsed by BMJ. Maps are provided without any warranty of any kind, either express or implied.

**Competing interests** AO, at the time of the study, is the Deputy Director General of the Ghana Health Service. The Ghana Health Service generates and owns the birth data analysed in this study and is responsible for healthcare delivery in Ghana. WD-G previously worked with the Ghana Health Service as a public health information officer until 2017.

**Patient and public involvement** Patients and/or the public were not involved in the design, or conduct, or reporting, or dissemination plans of this research.

**Patient consent for publication** Not required.

**Ethics approval** This study received ethical approval from the University of Southampton (ref: 54949.A1 and 54944) and the Ghana Health Service ethics review committee (ref: GHS-ERC008/05/20). Informed consent was obtained from all participants in the service provision assessment. All methods were carried out in accordance with relevant guidelines and regulations.

**Provenance and peer review** Not commissioned; externally peer reviewed.

**Data availability statement** Data are available in a public, open access repository. Data are available upon reasonable request. Data may be obtained from a third party and are not publicly available. The demographic dataset analysed during the current study is openly available in the WorldPop repository (https://www. worldpop.org/). The service provision assessment dataset analysed during the current study is available from the corresponding author on reasonable request. The birth datasets analysed during the current study are not publicly available due to confidentiality and data licensing restrictions from the Ghana Health Service. They can be obtained from the Ghana Health Service (https://www.ghs.gov.gh/contact-us) with reasonable request. Spatial data on roads and rivers are openly available

from OpenStreetMap for download through Geofabrik (https://www.geofabrik.de/). Landcover data are openly available from the European Space Agency (https://worldcover2020.esa.int/download).

**ORCID iD**
Winfred Dotse-Gborgbortsi http://orcid.org/0000-0001-7627-1809

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
