## [Reviewer comments · BMJ Open]

ARTICLE DETAILS

TITLE (PROVISIONAL)	Quality of maternal health care and travel time influence birthing service utilisation in Ghanaian health facilities: A geographic analysis of routine health data
AUTHORS	Dotse-Gborgbortsi, Winfred; Tatem, Andrew; Matthews, Zoe; Alegana, V; Ofosu, Anthony; Wright, Jim

VERSION 1 – REVIEW

REVIEWER	Liberty Makacha Midlands State University, Place Alert Labs, Surveying and Geomatics
REVIEW RETURNED	14-Sep-2022

GENERAL COMMENTS	The study uses GIS accessibility analysis to investigate the linkages or interactions between (1) Quality maternal health services and (2) travel time to affect birthing service utilization in Eastern region of Ghana. The manuscript make a very good read and presents an important and topical gap in modern LMIC health care utilization and determinants of health in Africa. However, the authors need to heavily copyedit the manuscript, as they seem to be making swift statements in their write-up. Some of the swift statements have negative connotations even on the choice of methods of analysis. Abstract: Generally, the abstract would require an improved writing style to capture the important messages which the authors intent to communicate. Line 37 - 39: "The birth records were from January to December 2017 obtained from the Ghana Health Service District Health Information Management Systems or manually extracted from hand written registers." Consider rephrasing to make the abstract statement more appealing to the reader. Proposed : Birth records for the period January to December 2017 were either manually from handwritten registers or obtained from the Ghana Health Service District Information Management System. Line 40: Consider breaking sentence into 2 parts. Line 40 - 42: This sentence carries 2 different important messages coming out of the manuscript and would sent across these messages if broken down into separate parts by message being conveyed. Line 44-48: Again consider rephrasing and/or breaking down sentence to capture important themes. At least a statement pointing to the sampling criteria for this study would help motivate for the strong statements on proportions affected in the results. Are these proportions not a reflection of sampling shortfalls in the study? Please motivate at least in the body of the study.
---

	Line 47 - 48: "The few CEmONC services were in urban centres, disadvantaging rural populations." To what extent does this statement take care of the boundary problem in studies of this nature. Line 67 - 68: Why are the authors confident in their statement that removing women whose places of residence could not be mapped would not lead to selection bias? I am of the opinion that this would be a source of bias for this study, though for the methods used in the study this component of participants could not be used in the analysis. Line 69 - 70: I am not convinced that accounting for individual level characteristics in aggregate models is not possible. Consider nested/hierarchical models. Introduction Line 84 -85: Are these gaps in research or health system gaps? While the statement is partially true, its presentation would somehow make it a wrong statement, at least in my opinion. Line 103: Research across LMICs rarely use(s) routine health data....., Check sentence construction. Line 106: Ones-off sub-national surveys not one-off. Line 111- 112: Consider rephrasing along "Quality of maternal healthcare impacts decisions to seek timely, adequate care by women at health facilities" Line 112 -116: Staff morale appears in the two parts to the message. Would this be implying different levels of meaning? Line 121: How does this compare with Ghana's healthcare facilities improvement and/or construction? Line 130 - 138: Why did the authors choose to address these three objectives in the specific order outlined here? Line 141: What is a spatial interaction analysis? A realized flow between an origin and a destination. It is a demand / supply relationship expressed over a geographical space? Maybe a lay statement here would help. Line 145 - 153: Are all CEmOC facilities from these statistics included in the spatial/location models? Line 164 - 166: This statement in my opinion would be contradictory to earlier assertions that individual level assessments were not possible from the available data. Authors may need to reconsider methods of assessment at least from a statistical point of view in light of the implications of this statement. Line 119 - 122: To what extent does the evidence in the literature on transit times to healthcare in Ghana tie in with developments on the service upgrade front? Line 130 - 138: I am wondering why the authors chose the specific sequence to answering the study objectives. Is this only in the wording of this paragraph or for some specific scientific reasons? I would not make the objectives sequentially addressed. Consider rephrasing. Results Line 277 - 281: To what extent are we powered to incorporate individual level modelling in this study using the 42205 geocoded records? Reporting statistics in the manuscript: Consider basic statistical reporting guidelines for articles published by BMJ e.g. decimal precision in reporting stats Line 354: The reference health facility is a health centre scoring 47%
--	--

	maternal health quality. Please justify this choice. Line 378 - 379: We note that for the setting of this study women stay beyond the WHO recommended 2hr travel time threshold. Maybe an equally important discussion point would be the extent to which the WHO threshold is realistic for this and most other African settings. Line 381 - 386: It looks like this work is benchmarked on only one previous study and has ignored what is in mainstream literature on the subject. While this is OK in improving previous work by the authors, there is a strong need to explore linkages and differences in findings between their work and what has happened in the research domain lately.
--	---

REVIEWER	J Nolan Northern Kentucky University, Mathematics & Statistics
REVIEW RETURNED	09-Oct-2022

GENERAL COMMENTS	This is a statistical review, however I will qualify this review by disclosing to the author and editor that I have no specific experience with spatial models and therefore am not able to evaluate the specifics of those models. I mark N/A to #8 above for this reason (the description does seem strong), and I would not be at all offended should the editor choose to reach out to a different statistical reviewer having that experience. SAMPLE: The study utilizes a sample of 40911 women (births) from 964 locations in 131 health facilities throughout Eastern Region, Ghana. This seems to be out of 1136 primary health facilities, authors should provide a better description of how the 131 facilities were chosen, as well as a description of what defines a location. Methods also mention that there were exclusions for various reasons. How many? And what if any bias seems likely/unlikely because of them? METHODS/RESULTS: First, I'm operating under the assumption that the statistical methods chosen here are reasonable. I have no reason to disbelieve that, but also lack the experience in this area of statistics that might be necessary to confirm it. Hence most of my comments address presentation of results. Most of the statistics discussed in results (lines 276-365) are descriptive in nature. Do the methods being used allow for inference (p-values, confidence intervals)? If so, this should be added; if not the limitations of making decisions based on descriptive statistics should be acknowledged. (For example, line 331 you say that a certain increase "doubled" the count of women giving birth and provide 208.3% as evidence. Without a confidence interval it is difficult for the reader to evaluate the strength of that evidence. In the next page you talk about a model R-square of 10.6%. That seems quite low, and lacking other inferential information hints at potential lack of relevance of the model. There are some confidence intervals given in Table 1, but their meaning is not really explained anywhere and they seem disconnected from the text. More clarity is desired in the results section. Overall, the presentation seems to me pretty well done. I saw no evidence in the paper to suggest that any of the methods used are wrong. I suggest addressing the items above, and sending it for review by someone with substantial experience in spatial statistics if desired.
--

REVIEWER	BL Solanke Obafemi Awolowo University, Demography and Social Statistics
REVIEW RETURNED	19-Oct-2022

GENERAL COMMENTS	The paper is well-written Adequately conceptualized and executed. However, I observed that in the modeling, the joint effects of travel time and quality of maternal healthcare on utilization of birthing service was not examined. The authors may wish to consider this to improve the analyzes.
--

VERSION 1 – AUTHOR RESPONSE

Reviewer: 1Dr. Liberty Makacha, Midlands State University, King's College London

Comments to the Author:

The study uses GIS accessibility analysis to investigate the linkages or interactions between (1) Quality maternal health services and (2) travel time to affect birthing service utilization in Eastern region of Ghana. The manuscript make a very good read and presents an important and topical gap in modern LMIC health care utilization and determinants of health in Africa. However, the authors need to heavily copyedit the manuscript, as they seem to be making swift statements in their write-up. Some of the swift statements have negative connotations even on the choice of methods of analysis.

Response: Thank you for taking time to revise the paper. We found your comments useful and they have improved our paper.

Abstract:

Generally, the abstract would require an improved writing style to capture the important messages which the authors intent to communicate.

Line 37 - 39: "The birth records were from January to December 2017 obtained from the Ghana Health Service District Health Information Management Systems or manually extracted from hand written registers." Consider rephrasing to make the abstract statement more appealing to the reader. Proposed : Birth records for the period January to December 2017 were either manually from handwritten registers or obtained from the Ghana Health Service District Information Management System.

Response: These suggestions have been implemented and manuscript revised accordingly. Thanks for the proposed rephrasing.

Line 40: Consider breaking sentence into 2 parts.

Response: The sentence has been separated into two.

Line 40 - 42: This sentence carries 2 different important messages coming out of the manuscript and would sent across these messages if broken down into separate parts by message being conveyed.

Response: We have revised this sentence.

Line 44-48: Again consider rephrasing and/or breaking down sentence to capture important themes. At least a statement pointing to the sampling criteria for this study would help motivate for the strong statements on proportions affected in the results. Are these proportions not a reflection of sampling shortfalls in the study? Please motivate at least in the body of the study.

Response: We have revised this sentence. There is no sampling issue related to the percentages reported in the abstract. The estimates of women within travel time groups were estimated with the WorldPop gridded population data, which is described under the "Demand population" sub-heading in the methods. However, we have added a sentence to the end of the "Travel time model" section of

the methods to remind readers that we used the WorldPop data. We also added the term 'census-derived' to the Methods section of the Abstract, thereby clarifying that these estimates are modelled from a full enumeration of population.

Line 47 - 48: "The few CEmONC services were in urban centres, disadvantaging rural populations." To what extent does this statement take care of the boundary problem in studies of this nature.

Response: The reference to "boundary problem" is not entirely clear. However, we assume this is referring to cross-border use of services as rural women would have to cross administrative boundaries and bypass other health facilities to use services in urban areas. We discussed bypassing and compared to other studies. The first paragraph of "Proximity effect and implications for maternal health" in the discussion section describes this phenomenon. The preceding sub-heading also mentioned it briefly.

Line 67 - 68: Why are the authors confident in their statement that removing women whose places of residence could not be mapped would not lead to selection bias? I am of the opinion that this would be a source of bias for this study, though for the methods used in the study this component of participants could not be used in the analysis.

Response: We do now admit that the exclusion of data could lead to some selection bias. This is reflected in our revised statement.

Line 69 - 70: I am not convinced that accounting for individual level characteristics in aggregate models is not possible. Consider nested/hierarchical models.

Response:

Given the pattern of available data and modelling framework, there are three reasons why we could not readily fit a multi-level model:

(a) Only a limited number of women's characteristics are recorded on patient registers (age, parity, level of education, occupation)

(b) These characteristics are only known for women who give birth in health facilities, not all women giving birth.

(c) The gravity spatial interaction model focuses on the mass at two locations (origin and destination) as well as the distance or friction between them to model flows. The form of models used does not include the characteristics of the people moving between the origin and destination.

(<https://www.sciencedirect.com/science/article/pii/B9780080449104007379>)

However, in a future study, having identified widespread by-passing via the analysis in our current manuscript, we recognise that it would be possible to use a multi-level model to analyse facility by-passing as an outcome in relation to facility-level characteristics and the limited individual-level characteristics that are recorded on patient registers. In view of this, we have added a sentence to the 'routine birth data' section of the 'Methods' to clarify what is recorded on patient registers. Based on the reviewer's suggestion, we also added some text to our Discussion to describe this potential avenue for follow-up analysis.

Introduction

Line 84 -85: Are these gaps in research or health system gaps? While the statement is partially true, its presentation would somehow make it a wrong statement, at least in my opinion.

Response: The initial sentence referred to maternal health system quality gaps. The sentence is revised to improve clarity.

Line 103: Research across LMICs rarely use(s) routine health data....., Check sentence construction.

Response: The sentence is revised to improve clarity.

Line 106: Ones-off sub-national surveys not one-off.

Response: We think our use and spelling of "one-off" is correct. We have not revised this section

Line 111- 112: Consider rephrasing along "Quality of maternal healthcare impacts decisions to seek timely, adequate care by women at health facilities"

Response: The sentence is revised to improve clarity.

Line 112 -116: Staff morale appears in the two parts to the message. Would this be implying different levels of meaning?

Response: The sentence is revised to improve clarity.

Line 121: How does this compare with Ghana's healthcare facilities improvement and/or construction?

Response: We have added a sentence to relate the universal health coverage initiative in Ghana to the quality of services.

Line 130 - 138: Why did the authors choose to address these three objectives in the specific order outlined here?

Response: There is no implied order here. If the words used are confusing, we have revised them.

Line 141: What is a spatial interaction analysis? A realized flow between an origin and a destination. It is a demand / supply relationship expressed over a geographical space? Maybe a lay statement here would help.

Response: Thanks for the suggestion. We have added a non-technical definition spatial interaction model.

Line 145 - 153: Are all CEmOC facilities from these statistics included in the spatial/location models?

Response: Yes. All the CEmONC facilities were included in the spatial interaction model.

Line 164 - 166: This statement in my opinion would be contradictory to earlier assertions that individual level assessments were not possible from the available data.

Authors may need to reconsider methods of assessment at least from a statistical point of view in light of the implications of this statement.

Response: The statement here describing the routine data does not imply that we had access to detailed attributes of women giving birth. The nature of the data and our statistical choice does not invalidate our approach. The individual level data was aggregated by origin and destination for the spatial interaction modelling as explained in response to similar comment above. Lines 166 to 167 indicate that we used the residential address and health facility for the gravity-type spatial interaction model. Indeed, these are the only data necessary data in the routine health records for a spatial interaction model. Therefore, we are unable to change our model as we used the right one. However, we have added a sentence to the conclusion to say we shall conduct a multilevel analysis to predict bypassing in the future. However, we have added the list of individual characteristics collected but not analysed at the end of the "routine birth data" methods section (lines 168 to 170).

Line 119 - 122: To what extent does the evidence in the literature on transit times to healthcare in Ghana tie in with developments on the service upgrade front?

Response: We have added a sentence to highlight the major initiative that scales up access to health care services in Ghana.

Line 130 - 138: I am wondering why the authors chose the specific sequence to answering the study objectives. Is this only in the wording of this paragraph or for some specific scientific reasons? I would not make the objectives sequentially addressed. Consider rephrasing.

Response: As mentioned earlier, there is no implied order here. If the words used are confusing, we have revised them.

Results

Line 277 - 281: To what extent are we powered to incorporate individual level modelling in this study using the 42205 geocoded records?

Response: As explained earlier, we are unable to implement an individual level modelling. The individual characteristics are not sufficient and the modelling framework does not incorporate individual characteristics of women moving between residential place and health facilities. If the comment is about statistical power, the study does not need to be powered. The data analysed is sufficiently high. Most studies would have selected a smaller sample, but we used all available data. Therefore, there was no need for powering a sample.

Reporting statistics in the manuscript: Consider basic statistical reporting guidelines for articles published by BMJ e.g. decimal precision in reporting stats

Response: We have gone through the generic BMJ guidelines (<https://www.bmj.com/about-bmj/resources-authors/article-types>) and found no specific requirement for decimal precision. Furthermore, we have sampled a few papers in the journal that reported their findings in similar fashion. We do not think our current presentation is inappropriate. Thus, we would make these changes if the editors ask us to do so or in the light of more specific recommendations from the reviewer.

Line 354: The reference health facility is a health centre scoring 47% maternal health quality. Please justify this choice.

Response: The phrasing of our sentence might be misleading. We have revised it for clarity. The reference health facility had the median quality score. The names of the health facilities were treated as factors in the destination constrained model. The quality scores were not included in this destination constrained model. The quality score was just reported here to give context when interpreting primary and secondary health facilities. This style was adopted as it would be needless to interpret all the 131 health facilities.

Line 378 - 379: We note that for the setting of this study women stay beyond the WHO recommended 2hr travel time threshold. Maybe an equally important discussion point would be the extent to which the WHO threshold is realistic for this and most other African settings.

Response: We discussed this recommended distance threshold (lines 467 to 477). We have added an extra sentence based on your comment. The recommendation is based on the fact that a woman who is bleeding would die within two hours if she does not receive the needed care such as blood transfusion. This recommendation is found on page 14 (Table 6) of the guideline for monitoring emergency obstetric care (https://www.unfpa.org/sites/default/files/pub-pdf/obstetric_monitoring.pdf). Also, in Sub-Saharan Africa, the leading direct cause (24.5%) of maternal deaths is haemorrhage. We emphasised the importance of the recommendation. Furthermore, we have added context on policy discussions about the distribution of specialist services from lines 470 to 473.

Line 381 - 386: It looks like this work is benchmarked on only one previous study and has ignored what is in mainstream literature on the subject. While this is OK in improving previous work by the authors, there is a strong need to explore linkages and differences in findings between their work and what has happened in the research domain lately.

Response: We have revised the second paragraph to highlight other studies. The second paragraph of the discussion as rightly pointed out describes the improvements on our previous analysis. Subsequently, the section on "Quality care effect and implications for maternal health" from lines 405 to 446 discussed our results with comparisons to other studies. The only difference is that the other studies did not use a similar modelling approach.

Reviewer: 2

Dr. J Nolan, Northern Kentucky University Comments to the Author:

1. Is the research question or study objective clearly defined?

Yes

2. Is the abstract accurate, balanced and complete?

Yes

3. Is the study design appropriate to answer the research question?

Yes

4. Are the methods described sufficiently to allow the study to be repeated?

Yes

5. Are research ethics (e.g. participant consent, ethics approval) addressed appropriately?

Yes

6. Are the outcomes clearly defined?

Yes

7. If statistics are used are they appropriate and described fully?

N/A

8. Are the references up-to-date and appropriate?

N/A

9. Do the results address the research question or objective?

Yes

10. Are they presented clearly?

Yes

11. Are the discussion and conclusions justified by the results N/A

12. Are the study limitations discussed adequately?

No

13. Is the supplementary reporting complete (e.g. trial registration; funding details; CONSORT, STROBE or PRISMA checklist)?

N/A

14. To the best of your knowledge is the paper free from concerns over publication ethics (e.g. plagiarism, redundant publication, undeclared conflicts of interest)?

N/A

15. Is the standard of written English acceptable for publication?

Yes

This is a statistical review, however I will qualify this review by disclosing to the author and editor that I have no specific experience with spatial models and therefore am not able to evaluate the specifics

of those models. I mark N/A to #8 above for this reason (the description does seem strong), and I would not be at all offended should the editor choose to reach out to a different statistical reviewer having that experience.

SAMPLE: The study utilizes a sample of 40911 women (births) from 964 locations in 131 health facilities throughout Eastern Region, Ghana. This seems to be out of 1136 primary health facilities, authors should provide a better description of how the 131 facilities were chosen, as well as a description of what defines a location. Methods also mention that there were exclusions for various reasons. How many? And what if any bias seems likely/unlikely because of them?

Response: We have added a sentence for clarity (lines 188 to 190). As this new sentence makes clear, only a minority of the 1136 health facilities in the study area provide birthing services, so most were ineligible for the study. We also clarify that we could not get the routine individual-level health records in 19 health facilities. We have reported the bias relating to the 19 health facilities on lines 490 to 492. Also, we reviewed all uses of the term 'locations', changing this to 'residential locations' or 'facility locations' to clarify our meaning.

METHODS/RESULTS: First, I'm operating under the assumption that the statistical methods chosen here are reasonable. I have no reason to disbelieve that, but also lack the experience in this area of statistics that might be necessary to confirm it. Hence most of my comments address presentation of results.

Most of the statistics discussed in results (lines 276-365) are descriptive in nature. Do the methods being used allow for inference (p-values, confidence intervals)? If so, this should be added; if not the limitations of making decisions based on descriptive statistics should be acknowledged. (For example, line 331 you say that a certain increase "doubled" the count of women giving birth and provide 208.3% as evidence. Without a confidence interval it is difficult for the reader to evaluate the strength of that evidence. In the next page you talk about a model R-square of 10.6%. That seems quite low, and lacking other inferential information hints at potential lack of relevance of the model. There are some confidence intervals given in Table 1, but their meaning is not really explained anywhere and they seem disconnected from the text. More clarity is desired in the results section.

Response The coefficients were standardised and presented as percentage. We have now linked those percentages to the corresponding incidence rate ratio with confidence intervals. All the results in the subsection "Evaluating the influences on birthing service utilisation" are inferential, whereas those in the preceding section are descriptive and based either on patient registers or census-derived gridded population layers. We have added Confidence Intervals to statistics in text where possible throughout this sub-section. We have not added the pvalues because they can be inferred from the confidence intervals. Furthermore, the confidence intervals also better presents the uncertainties related to our estimates. We have now specified in the methods (lines 266 and 267) and results (lines 331 and 332) that standardised coefficients are used as added interpretation approach.

Overall, the presentation seems to me pretty well done. I saw no evidence in the paper to suggest that any of the methods used are wrong. I suggest addressing the items above, and sending it for review by someone with substantial experience in spatial statistics if desired.

Response: Thanks for your kind comments.

Reviewer: 3

Dr. BL Solanke, Obafemi Awolowo University Comments to the Author:

The paper is well-written

Adequately conceptualized and executed. However, I observed that in the modeling, the joint effects of travel time and quality of maternal healthcare on utilization of birthing service was not examined. The authors may wish to consider this to improve the analyzes.

Response: Thanks for your comments. We have implemented your suggestion to estimate the joint effect of travel time and quality on utilisation of services. See new figure 3D and intext description from lines 360 to 363.

VERSION 2 – REVIEW

REVIEWER	Liberty Makacha Midlands State University, Place Alert Labs, Surveying and Geomatics
REVIEW RETURNED	09-Dec-2022

GENERAL COMMENTS	The authors did consider recommendations given by the two reviewers. Critically appraised justifications and reasons to depart from proposals were also given where frames of references differed. The manuscript in its revised state is an excellent contribution to the body of knowledge.
--

REVIEWER	BL Solanke Obafemi Awolowo University, Demography and Social Statistics
REVIEW RETURNED	27-Dec-2022

GENERAL COMMENTS	The authors have adequately addressed the queries. The paper is suitable for publication.
---